# Research on a Rail Defect Location Method Based on a Single Mode Extraction Algorithm

**Bo Xing** [1], **Zujun Yu** [1,2], **Xining Xu** [1,2,*], **Liqiang Zhu** [1,2] and **Hongmei Shi** [1,2]

1   School of Mechanical, Electronic and Control Engineering, Beijing Jiaotong University, Beijing 100044, China; 15116333@bjtu.edu.cn (B.X.); zjyu@bjtu.edu.cn (Z.Y.); lqzhu@bjtu.edu.cn (L.Z.); hmshi@bjtu.edu.cn (H.S.)
2   Key Laboratory of Vehicle Advanced Manufacturing, Measuring and Control Technology (Beijing Jiaotong University), Ministry of Education, Beijing 100044, China
*   Correspondence: xnxu@bjtu.edu.cn

**Abstract:** This paper proposes a rail defect location method based on a single mode extraction algorithm (SMEA) of ultrasonic guided waves. Simulation analysis and verification were conducted. The dispersion curves of a CHN60 rail were obtained using the semi-analytical finite element method, and the modal data of the guided waves were determined. According to the inverse transformation of the excitation response algorithm, modal identification under low-frequency and high-frequency excitation was realized, and the vibration displacements at other positions of a rail were successfully predicted. Furthermore, an SMEA for guided waves is proposed, through which the single extraction results of four modes were successfully obtained when the rail was excited along different excitation directions at a frequency of 200 Hz. In addition, the SMEA was applied to defect location detection, and the single reflection mode waveform of the defect was extracted. Based on the group velocity of the mode and its propagation time, the distance between the defect and the excitation point was measured, and the defect location was predicted as a result. Moreover, the SMEA was applied to locate the railhead defect. The detection mode, the frequency, and the excitation method Were selected through the dispersion curves and modal identification results, and a series of signals of the sampling nodes were obtained using the three-dimensional finite element software ANSYS. The distance between the defect and the excitation point was calculated using the SMEA result. When compared with the structure of the simulated model, the errors obtained were all less than 0.5 m, proving the efficacy of this method in precisely locating rail defects, thus providing an innovated solution for rail defect location.

**Keywords:** rail; ultrasonic guided wave; semi-analytical finite element; single mode extraction algorithm; defect location

## 1. Introduction

Ultrasound guided waves are widely used in the non-destructive testing of continuous welded rails because of their wide coverage and rapid propagation over large distances [1,2]. Compared with simple waveguide structures, such as plates and pipelines, rails have much more complex cross-sectional structures; thus, they require far more complicated ultrasonic wave modes for defect detection. With the increase in the frequency, the number of modes increases, making the research more difficult. However, the analytical method is no longer suitable for the analysis of the propagation characteristics of ultrasonic guided waves in rails. Therefore, numerical analysis methods have been broadly introduced, including the Boundary Element Method [3], the Finite Difference Method [4], the Finite Element Method [5], and the Semi-Analytical Finite Element Method [6].



Among them, the semi-analytical finite element method reduces the analysis dimensions and has a higher computational efficiency, thus is commonly adopted in this field of researches.

The time difference between the main wave packet of the reflected wave and the excitation wave packet in the acquisition signal must be calculated; the position of the defect can be inferred by multiplying the group velocity of the main modes. Its accuracy is related to the selected detection mode. Therefore, it is necessary to have a good understanding of the propagation characteristics of guided waves in rails to select the appropriate modal type, frequency, and excitation mode for defect detection. D. Alleyne and P. Cawley [7] used two-dimensional Fourier transform to separate the guided wave modes and calculate the corresponding phase velocities.T. Hayashi et al. [8] utilized the semi-analytical finite element (SAFE) method to calculate the group velocity and phase velocity dispersion curves of guided waves in rails at frequencies of 0–100 kHz and carried out experimental verification. He Cunfu et al. [9] used the vibration mode analysis method to analyze the guided wave modes at frequencies of 0–50 kHz; the dispersion characteristics and wave structures of five typical modes were analyzed, and the mode types and frequency ranges suitable for inspecting railhead and rail bottom defects were obtained. P.W. Loveday and C.S. Long [10–14] analyzed the guided wave mode data in rails at frequencies of 25 kHz and 35 kHz using the SAFE method and measured along the rails using a laser vibrometer. A large number of vibration data were obtained, the amplitudes of each mode were calculated, and the vibration waveforms of distant rails were successfully predicted. In 2017, P.W. Loveday et al. [15] studied the problem of the mode repulsion and crossing behavior of the approaching wave number versus frequency curves by analyzing the second derivative of the eigenvalue with respect to the wave number. These methods analyze the modal characteristics of guided waves propagating in rails, giving a modal analysis. However, it has never been possible to separate the reflected guided wave modes of defects to use them for defect location directly.

Meanwhile, a large number of research teams have applied ultrasonic guided waves for defect detection in waveguide structures. Most of the research results focus on simple cross-sectional structures such as plates and pipes, while studies on complex structures such as rails remain relatively rare. C.M. Lee et al. [2] adopted the finite element method to analyze the energy distribution characteristics of guided waves with different frequencies in rails. The simulation results showed that a high-frequency guided wave of 200 kHz is concentrated on the upper surface of the railhead, and a low-frequency guided wave of 30 kHz is distributed through the entire railhead. The guided waves at low frequencies are more sensitive to the transverse cracks of the railhead, while the guided waves at high frequencies are good at detecting defects in the shelling. These results have been experimentally verified. Lu Chao et al. [16] selected transverse and vertical vibration modes to detect oblique cracks on the rail bottom and analyzed the relationship between the angle of the oblique cracks and the scattering characteristics of the guided waves. G. Zumpano et al. [17] utilized finite element software to simulate the rail wear and applied different excitation frequencies. Location analysis shows that the location error is affected by the excitation frequency. However, as a result of the dispersion characteristics of ultrasonic guided waves and the frequency aliasing phenomenon, it is impossible to accurately locate the defect from the group velocity. For this reason, there are relatively few studies on the application of ultrasonic guided waves in rail defects location. Nevertheless, because of the results related with the prediction of defect sizes and the identification of defect types, it seems practical to study this method in terms of defect location.

To locate rail cracks accurately by using ultrasonic guided waves, it is generally necessary to obtain a single mode propagating in the rail. The research ideas can be roughly divided into two kinds: one is to directly excite a single mode in the rail, and the other is to extract a single mode from the signal propagating in the rail. In previous studies, the research group proposed a single mode excitation method, which can excite a relatively pure single mode in rails [18]. At the same time, the group also carried out research on the optimal mode selection of rail crack detection in which a selection model of crack detection mode is created. For a specific crack, the guided wave frequency and mode suitable for crack detection are quickly selected. In this paper, a single mode

extraction algorithm (SMEA) is proposed to ascertain the precise location of defects. The method is as follows: Firstly, the mode and frequency of defect detection are selected and a three-dimensional model of the defect present in the railhead is established. The simulation analysis is carried out using the three-dimensional finite element analysis software ANSYS. The defect models of the railhead are stimulated with low-frequency (200 Hz) and high-frequency (60 kHz) signals. The content of each mode of guided wave is quantitatively analyzed, and the position of the railhead defect is calculated according to the single mode extraction method. The semi-analytical finite element method and accurate modal identification method are described in Section 2. In Section 3, the SMEA and verification process are discussed. The defect location method and simulation results are described in Section 4. Conclusions are given in Section 5.

## 2. An Accurate Modal Identification Method

### 2.1. Basic Characteristics of Ultrasonic Guided Waves in Rails

As a result of the complex cross-sectional structure of rails, the number of guided wave modes propagating in a rail is large. To detect the internal defects of rails based on ultrasonic guided wave technology, the most important thing is to grasp the fundamental characteristics of ultrasonic guided waves in rails, such as the frequency, wave number, phase velocity, group velocity, mode shape, and other information, so as to analyze the propagation characteristics of the guided waves. The dispersion curves of ultrasonic guided waves in rails can be obtained with the semi-analytical finite element method. Taking the rail laid on Beijing–Shanghai high-speed railway in China as the research object [19], the finite element method was used to discretize the cross-section. It was assumed that the guided waves propagate along the longitudinal direction of the rail in the form of harmonic vibrations. The wave equation was established based on the finite element method. The eigenvalues and eigenvectors were obtained by solving the eigenvalue equation. The eigenvalues contain the information of frequencies and wave numbers, and the eigenvectors contain the information of mode shapes. Thus, the dispersion curves and mode shapes of guided waves in the CHN60 rail were obtained.

First, the coordinate system of the CHN60 rail was established, as shown in Figure 1.

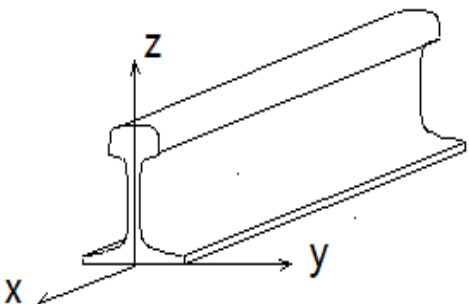

**Figure 1.** CHN60 rail coordinate system.

The wave number of guided waves is $\xi$ and the frequency is $\omega$. The displacement, stress, and strain of each node in the rail can be expressed as Equation (1).

$$u = \begin{bmatrix} u_x & u_y & u_z \end{bmatrix}^T$$

$$\sigma = \begin{bmatrix} \sigma_x & \sigma_y & \sigma_z & \sigma_{yz} & \sigma_{xz} & \sigma_{xy} \end{bmatrix}^T \tag{1}$$

$$\varepsilon = \begin{bmatrix} \varepsilon_x & \varepsilon_y & \varepsilon_z & \gamma_{yz} & \gamma_{xz} & \gamma_{xy} \end{bmatrix}^T$$

$\varepsilon_x$, $\varepsilon_y$, and $\varepsilon_z$ are normal strains, and $\gamma_{yz}$, $\gamma_{xz}$, and $\gamma_{xy}$ are shear strains. In the elastic range of materials, stress and strain conform to Hooke's Law. It can be expressed as $\sigma = C\varepsilon$, where $C$ is the elastic constant matrix of a rail.

The relationship between the strain and displacement at any node in the rail is expressed in a matrix form as follows:

$$\varepsilon = \begin{bmatrix} L_x \frac{\partial u}{\partial x} + L_y \frac{\partial u}{\partial y} + L_z \frac{\partial u}{\partial z} \end{bmatrix} \tag{2}$$

Therefore, in the form

$$L_x = \begin{bmatrix} 1 & 0 & 0 \\ 0 & 0 & 0 \\ 0 & 0 & 0 \\ 0 & 0 & 0 \\ 0 & 0 & 1 \\ 0 & 1 & 0 \end{bmatrix} L_y = \begin{bmatrix} 0 & 0 & 0 \\ 0 & 1 & 0 \\ 0 & 0 & 0 \\ 0 & 0 & 1 \\ 0 & 0 & 0 \\ 1 & 0 & 0 \end{bmatrix} L_z = \begin{bmatrix} 0 & 0 & 0 \\ 0 & 0 & 0 \\ 0 & 0 & 1 \\ 0 & 1 & 0 \\ 1 & 0 & 0 \\ 0 & 0 & 0 \end{bmatrix} \tag{3}$$

The SAFE method was used to get the dispersion curves of the ultrasonic guided waves which propagate in the form of harmonics in the longitudinal direction of the rail. Therefore, only the finite element discretization of the rail cross-section was needed. The displacement of any discrete node in the rail can be calculated as shown in Equation (4), where $x$ is the longitudinal coordinate of the rail.

$$u(x,y,z,t) = \begin{bmatrix} u_x(x,y,z,t) \\ u_y(x,y,z,t) \\ u_z(x,y,z,t) \end{bmatrix} = \begin{bmatrix} U_x(y,z) \\ U_y(y,z) \\ U_z(y,z) \end{bmatrix} e^{-i(\xi x - \omega t)} \tag{4}$$

The triangular element was selected to discretize the cross-section of the CHN60 rail, and 177 nodes and 255 elements were obtained, as shown in Figure 2. Seven nodes are circled in the figure to illustrate the signal extraction nodes in the subsequent modal identification method.

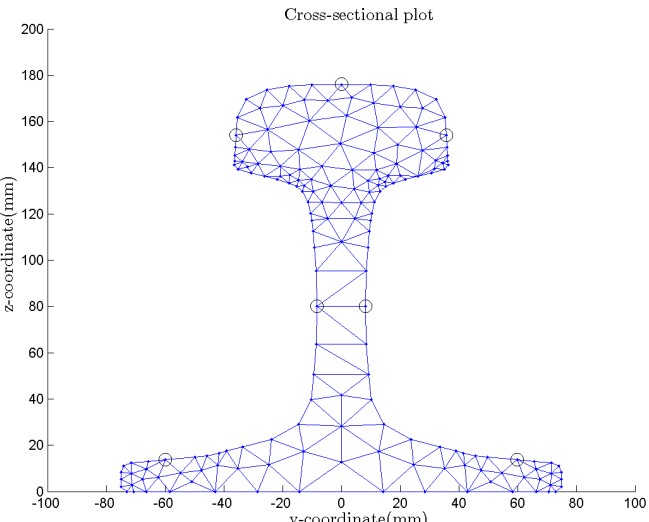

**Figure 2.** Discretization of the cross-section of the CHN60 rail.

The rail cross-section was discretized by triangular elements. The displacement of any particle in an element can be obtained by the displacement of the nodes and the shape function. The strain and

stress vectors of the element can also be expressed by the displacement of the nodes. According to the Hamilton principle, the dynamic equation of guided waves propagating in a CHN60 rail can be obtained by calculating the strain energy and potential energy at any point simultaneously [20]:

$$\left[ K_1 + i\xi K_2 + \xi^2 K_3 - \omega^2 M \right]_M U = 0 \tag{5}$$

In this form, $U$ contains the displacements of the nodes in $x$, $y$, and $z$ directions. $M$ is the mass matrix of the nodes. $\xi$ and $\omega$ are the wave number and angular frequency, respectively, and $K_1$, $K_2$, and $K_3$ are the stiffness matrices.

Given the frequency $\omega$, the eigenvalue of Equation (5) is the wave number $\xi$ of the guided waves, and the eigenvector contains the mode shapes of the rail. Normalized processing can be used to plot the corresponding mode shapes of the guided waves. The modes with pure imaginary or complex wave numbers are not considered here because these modes will exponentially decay as the distance increases and cannot propagate. The dispersion curves of the phase velocity and group velocity of ultrasonic guided waves in the CHN60 rail are plotted, as shown in Figure 3.

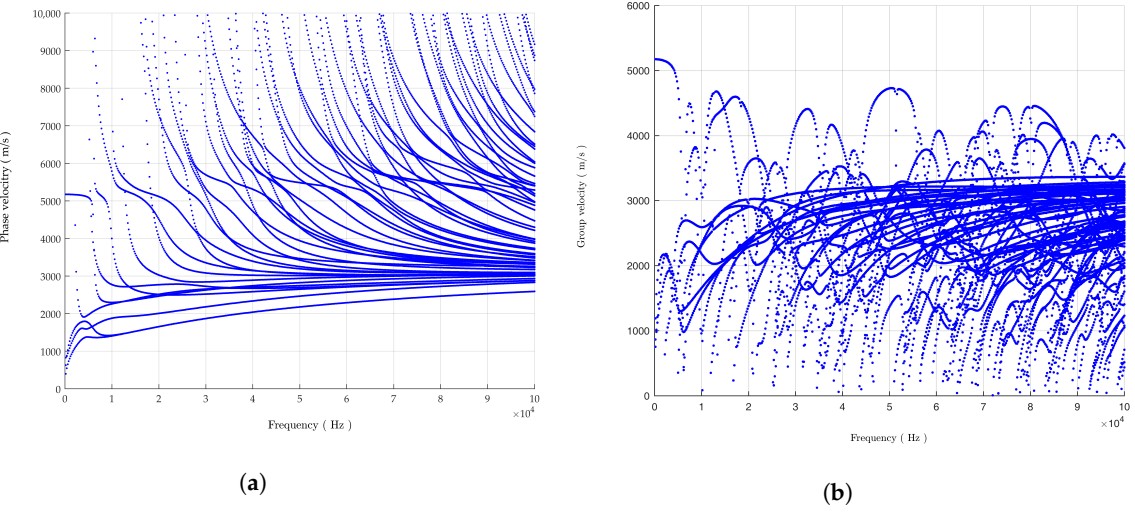

(**a**)                                       (**b**)

**Figure 3.** (**a**) Phase velocity; and (**b**) group velocity dispersion curves.

The dispersion curves of the phase velocity and group velocity are shown in the figure. At the same frequency, the rail has several propagable guided wave modes, and at the higher the frequency, it has more guided wave modes.

*2.2. Excitation Response Analysis of Rails*

On the basis of the excitation response analysis method, the vibration signals of a non-defective rail can be calculated at any point. Firstly, the system function model $U$ of the rail should be obtained. Then, the frequency domain signal $\widehat{F}(f)$ of the excitation function signal $v_1(t)$ can be obtained. Thereafter, the excitation response result can be calculated $V = U \times \widehat{F}(f)$. Finally, the inverse Fourier transform of $V$ can be determined, which is the excitation response of the time domain vibration signal.

The system function model of the rail [21] is shown in Equation (6), where $m$ is the mode number ($m = 1, 2, \ldots, M$). $\Omega_m$ is the amplitude and phase parameters of mode $m$. $U_m^L$ and $U_m^R$ are the left eigenvector and the right eigenvector of mode $m$. $\widetilde{p}$ is the amplitude of the excitation signal. $x_s$ and $x$ are the longitudinal coordinates of the excitation point and the receiving point, and $U_m^{Rup}$ is the mode shape of mode $m$.

$$U(y, z, f) = \sum_{m=1}^{M} \Omega_m U_m^{Rup} e^{-i[\xi_m(x - x_s)]} \tag{6}$$

In this form, the parameters are expressed as follow:

$$\Omega_m = -\frac{U_m^L \widetilde{p}}{B_m}$$

$$B_m = U_m^L B U_m^R$$

$$B = \begin{bmatrix} K_1 - \omega^2 M & 0 \\ 0 & -K_3 \end{bmatrix}$$

The result of the frequency response analysis of the excitation response $V(f)$ can be expressed by Equation (7).

$$V(y,z,f) = \widehat{F}(f) \cdot U(y,z,f)$$

$$= \widehat{F}(f) \cdot \sum_{m=1}^{M} \Omega_m U_m^{Rup} e^{-i[\xi_m(x-x_s)]} \tag{7}$$

The inverse Fourier transform result is the time domain vibration signal of the node.

### 2.3. Modal Identification

In Figure 4, T is the transmitter of the guided waves, R is the receiver of guided waves, X is the location of a rail defect, and the arrows show the propagation paths of the guided waves. Path 1 is the direction of backward propagation after exciting the guided wave. Paths 2 and 3 are the directions of transmission and reflection after the guided wave meets the crack. According to the propagation time of the reflected wave (Path 3) and the group velocity of the main modes, the guided wave propagating distance of Path 3 can be calculated to locate the defect.

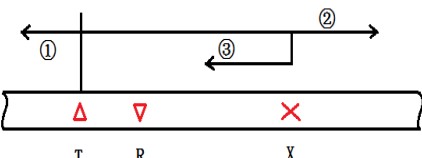

**Figure 4.** Schematic diagram of defect location.

To solve the problem of defect location, it is very important to analyze the mode propagation and reflection of guided waves in defective rails. The amplitude of each mode in a complete rail can be calculated using the excitation response algorithm, as described in Section 2.2, but the mechanisms involved in the interaction between modes and defects are still difficult to understand. Therefore, a method that can accurately analyze the direct mode and reflection mode of a defect is needed. In fact, the accurate modal identification method used in this paper is the inverse transformation of the excitation response analysis method.

### 2.3.1. Theoretical Derivation

According to the excitation response analysis method, the vibration displacement of any point of the rail is equal to the superposition of all the propagable modes' vibration displacements at that point. However, when there are both direct and reflected waves in the rail, the vibration displacement at any point is the superposition of the direct and reflected waves, that is, the superposition of all direct modes and all reflected modes at that point. Therefore, when considering the influence of the reflection wave caused by the defect, the expression of the excitation response of the node is in the form of Equation (8), in which the upper line "-" represents the relevant parameters of the reflected modes.

$$V(y, z, f) = \widehat{F}(f) \cdot U(y, z, f)$$

$$= \widehat{F}(f) \cdot \left( \sum_{m=1}^{M} \Omega_m U_m^{Rup} e^{-i[\xi_m(x-x_s)]} + \sum_{m=1}^{M} \bar{\Omega}_m \bar{U}_m^{Rup} e^{-i[\bar{\xi}_m(x-x_s)]} \right) \tag{8}$$

If we let $\Omega = \begin{bmatrix} \Omega_m & \bar{\Omega}_m \end{bmatrix}$, then with to the sampling results of a simulation or an experiment, the frequency domain signal $V(f)$ of any $N$ points' vibration displacement on the rail can be obtained. By substituting the wave number, the mode shape, the distance between the sampling point and excitation point, and the frequency domain signals of the excitation signal at the corresponding frequency of each mode into Equation (8), the parameters $\Omega$ including the amplitude and phase of each mode can be achieved. It should be noted that the number of sampling points should be enough to ensure that the solution can be obtained relatively accurately, that is, $N > 2M$.

### 2.3.2. Simulation Analysis

Due to the excessive modes of high frequency guided waves, the results of modal identification are complex and difficult to study. Therefore, in the process of research, we first judged the correctness of the algorithm by studying 200 Hz low-frequency guided waves, and then increased the frequency to ultrasound band to study the modal identification results of high-frequency guided waves.

Firstly, the rail model with a railhead defect was established, and the excitation response of the rail was simulated using ANSYS. The rail was then excited by signals at a low center frequency of 200 Hz and a high center frequency of 60 kHz. The rail and defect models are shown in Figure 5.

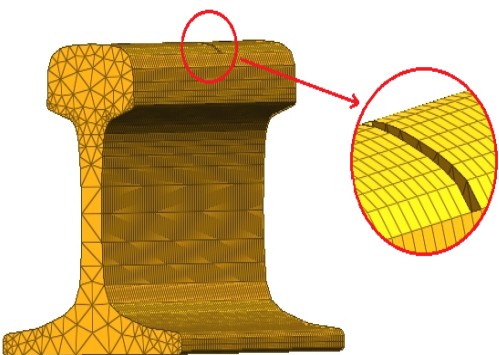

**Figure 5.** Rail model with head defect.

(1) Low center frequency of 200 Hz.

The length of the rail model was 200 m, and the defect was located at 120 m and had a length of 70 mm, width of 50 mm, and depth of 20 mm. The excitation signal was a sinusoidal wave modulated using Hanning window, with a center frequency of 200 Hz and five cycles. The excitation position was the center point of the side of the railhead at 90 m. A total of 32 sections were sampled from 100 m to 101.55 m. Seven nodes were selected for each section, as shown in Figure 2.

At 200 Hz, there were only four modes in the rail: the horizontal bending mode, the vertical bending mode, the torsional mode, and the extensional mode. The collected signals were substituted into the modal identification in Equation (8), which is extended into the form of Equation (9).

$$\widehat{F}(f) \begin{bmatrix} U_{1,1}^{Rup}e^{-i[\xi_1(x_1-x_S)]} & \cdots & U_{1,4}^{Rup}e^{-i[\xi_4(x_1-x_S)]} & \bar{U}_{1,1}^{Rup}e^{-i[\xi_1(x_1-x_S)]} & \cdots & \bar{U}_{1,4}^{Rup}e^{-i[\xi_4(x_1-x_S)]} \\ U_{1,1}^{Rup}e^{-i[\xi_1(x_2-x_S)]} & \cdots & U_{1,4}^{Rup}e^{-i[\xi_4(x_2-x_S)]} & \bar{U}_{1,1}^{Rup}e^{-i[\xi_1(x_2-x_S)]} & \cdots & \bar{U}_{1,4}^{Rup}e^{-i[\xi_4(x_2-x_S)]} \\ \vdots & & \vdots & \vdots & & \vdots \\ U_{1,1}^{Rup}e^{-i[\xi_1(x_{32}-x_S)]} & \cdots & U_{1,4}^{Rup}e^{-i[\xi_4(x_{32}-x_S)]} & \bar{U}_{1,1}^{Rup}e^{-i[\xi_1(x_{32}-x_S)]} & \cdots & \bar{U}_{1,4}^{Rup}e^{-i[\xi_4(x_{32}-x_S)]} \\ \vdots & & \vdots & \vdots & & \vdots \\ U_{7,1}^{Rup}e^{-i[\xi_1(x_1-x_S)]} & \cdots & U_{7,4}^{Rup}e^{-i[\xi_4(x_1-x_S)]} & \bar{U}_{7,1}^{Rup}e^{-i[\xi_1(x_1-x_S)]} & \cdots & \bar{U}_{7,4}^{Rup}e^{-i[\xi_4(x_1-x_S)]} \\ U_{7,1}^{Rup}e^{-i[\xi_1(x_2-x_S)]} & \cdots & U_{7,4}^{Rup}e^{-i[\xi_4(x_2-x_S)]} & \bar{U}_{7,1}^{Rup}e^{-i[\xi_1(x_2-x_S)]} & \cdots & \bar{U}_{7,4}^{Rup}e^{-i[\xi_4(x_2-x_S)]} \\ \vdots & & \vdots & \vdots & & \vdots \\ U_{7,1}^{Rup}e^{-i[\xi_1(x_{32}-x_S)]} & \cdots & U_{7,4}^{Rup}e^{-i[\xi_4(x_{32}-x_S)]} & \bar{U}_{7,1}^{Rup}e^{-i[\xi_1(x_{32}-x_S)]} & \cdots & \bar{U}_{7,4}^{Rup}e^{-i[\xi_4(x_{32}-x_S)]} \end{bmatrix} \begin{bmatrix} \Omega_1 \\ \Omega_2 \\ \vdots \\ \Omega_4 \\ \bar{\Omega}_1 \\ \bar{\Omega}_2 \\ \vdots \\ \bar{\Omega}_4 \end{bmatrix} = \begin{bmatrix} V_{1,1}(f) \\ V_{1,2}(f) \\ \vdots \\ V_{1,32}(f) \\ \vdots \\ V_{7,1}(f) \\ V_{7,2}(f) \\ \vdots \\ V_{7,32}(f) \end{bmatrix} \tag{9}$$

The magnitudes of each mode and the corresponding reflection mode (i.e., the modulus value of $\Omega$) and the amplitude reflection coefficients after calculation are shown in Table 1.

**Table 1.** Vertical excitation mode identification results of defective rail (at a frequency of 200 Hz).

| Mode | Direct Wave | Reflected Wave | Amplitude Reflection Coefficient |
|---|---|---|---|
| Horizontal bending mode | $3.20 \times 10^{-3}$ | $3.04 \times 10^{-4}$ | $9.5 \times 10^{-2}$ |
| Vertical bending mode | $1.89 \times 10^{-2}$ | $3.52 \times 10^{-3}$ | 0.19 |
| Torsional mode | $1.17 \times 10^{-2}$ | $4.58 \times 10^{-4}$ | $3.9 \times 10^{-2}$ |
| Extensional mode | $4.94 \times 10^{-4}$ | $5.95 \times 10^{-4}$ | 1.2 |

It shows that the vertical bending mode had the highest amplitude among the direct modes, which was the main mode to be stimulated in this way. The amplitude of the extensional mode was far smaller than the amplitudes of the other modes and could be seen as zero, thus it was not considered. Meanwhile, the amplitude reflection coefficient was obtained by dividing the amplitude of the reflected mode from that of the direct mode, and the vertical bending mode was the most sensitive mode to the transverse crack of the railhead. The results of the modal identification algorithm were used to estimate the vibration of the railhead at a distance of 20 m away from the excitation point and were compared with the simulation sampling results at the same point, as shown in Figure 6a. The two methods' results almost coincide, which proves the validity of the algorithm in the modal identification of guided waves at a low frequency.

(2) High center frequency of 60 kHz

The length of the rail model was 25 m, and the defect was located at 20 m, with a length of 70 mm, a width of 50 mm, and a depth of 20 mm. A 10-cycle sinusoidal wave with a center frequency of 60 kHz was applied to excite the railhead at 12 m vertically, and 32 sections were sampled between 12.3 m and 13.23 m. The nodes were selected as above.

At 60 kHz, there were 35 modes in the rail, thus it was impossible to simply distinguish the types of modes. For this reason, they are numbered in the order of phase velocity from small to large. Using the signal with a center frequency of 60 kHz to excite the railhead, the magnitude and reflection coefficients of each mode and the corresponding reflection modes were obtained (as shown in Table 2), after the application of the accurate modal identification method.

In Table 2, it can be seen that, among the direct modes, mode No. 7, No. 8, and No. 9 had relatively high amplitudes, as they were the main modes stimulated in this way. At the same time, in the reflection mode, mode No. 7, with the largest amplitude and the highest reflection coefficient, was the most sensitive mode to the transverse crack of the railhead. Similarly, the vibration of the railhead 1.5 m away from the excitation point was estimated using the calculation results in Table 2, and compared with the simulation sampling results at that point, as shown in Figure 6b. The two main wave packets almost coincide, which shows that the algorithm is also applicable to the modal identification of guided waves at a high frequency.

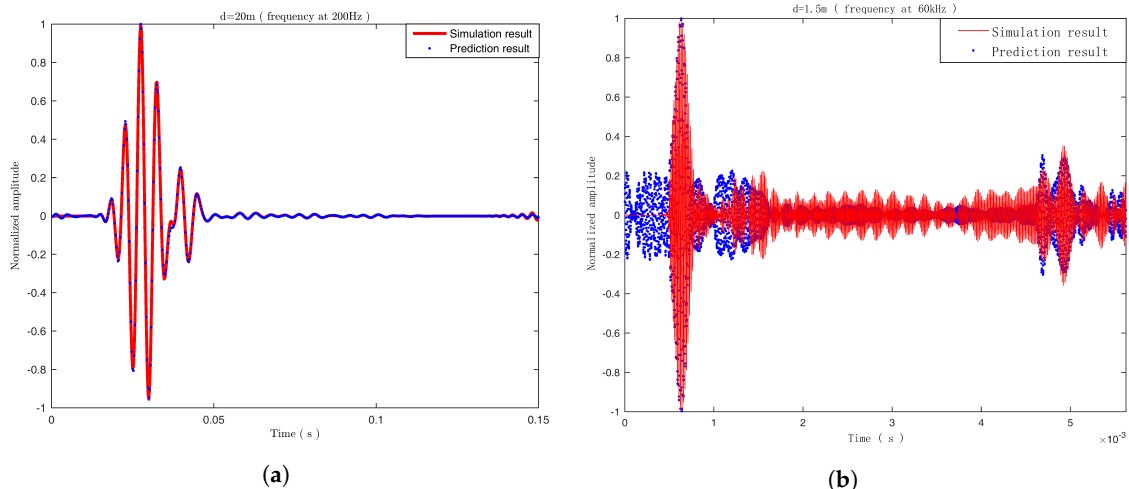

(**a**)                                                    (**b**)

**Figure 6.** Comparison between the simulation results and prediction results: (**a**) 200 Hz; and (**b**) 60 kHz.

**Table 2.** Results of accurate modal identification (at a frequency of 60 kHz).

| Mode Number | Direct Wave | Reflected Wave | Amplitude Reflection Coefficient |
|---|---|---|---|
| 1 | 0.39 | 0.003 | 0.008 |
| 2 | 0.18 | 0.009 | 0.050 |
| 3 | 0.03 | 0.001 | 0.033 |
| 4 | 0.29 | 0.007 | 0.024 |
| 5 | 0.33 | 0.003 | 0.009 |
| 6 | 0.21 | 0.008 | 0.038 |
| 7 | 1.25 | 0.078 | 0.062 |
| 8 | 1.46 | 0.004 | 0.003 |
| 9 | 1.36 | 0.004 | 0.003 |
| 10 | 0.52 | 0.002 | 0.004 |
| 11 | 0.08 | 0.001 | 0.013 |
| 12 | 0.23 | 0.002 | 0.009 |
| 13 | 0.40 | 0.003 | 0.043 |
| 14 | 0.67 | 0.001 | 0.001 |
| 15 | 0.07 | 0.003 | 0.043 |
| 16 | 0.06 | 0.001 | 0.017 |
| 17 | 0.96 | 0.005 | 0.005 |
| 18 | 0.04 | 0 | 0 |
| 19 | 0.37 | 0.006 | 0.016 |
| 20 | 0.02 | 0.001 | 0.050 |
| 21 | 0.29 | 0.005 | 0.017 |
| 22 | 0.15 | 0.002 | 0.013 |
| 23 | 0.06 | 0.001 | 0.017 |
| 24 | 0.16 | 0.003 | 0.019 |
| 25 | 0.23 | 0.008 | 0.035 |
| 26 | 0.28 | 0.001 | 0.004 |
| 27 | 0.36 | 0.002 | 0.006 |
| 28 | 0.41 | 0.001 | 0.002 |
| 29 | 0.42 | 0.002 | 0.005 |
| 30 | 0.03 | 0.001 | 0.033 |
| 31 | 0.32 | 0 | 0 |
| 32 | 0.42 | 0 | 0 |
| 33 | 0.92 | 0 | 0 |
| 34 | 0.02 | 0 | 0 |
| 35 | 0.11 | 0 | 0 |

### 3. Single Modal Extraction Algorithm

When ultrasonic guided waves propagate in rails, dispersion occurs. That is to say, the number of the guided wave modes propagating in the rails will increase with the increase in excitation frequency; furthermore, the propagating velocity of each mode is different. With the increase in propagating distance, the wave packets of each mode will gradually stagger and overlap. Therefore, they cannot be distinguished, and so the group velocity cannot be used to locate defects. To solve this problem, an SMEA based on the accurate modal identification method is proposed.

As mentioned above, the vibration displacement at any point on the rail is the superposition of the vibration displacement of all modes at this point. Conversely, the vibration displacement of each mode at any point can be split by the total displacement of the point.

The amplitude and phase $\Omega_n$ of mode $n$ can be obtained with the accurate modal identification algorithm, and the vibration displacement generated by mode $n$ at the longitudinal coordinate $x$ can be expressed by Equation (10).

$$V_n(y, z, f) = \bar{F}_f \cdot \Omega_n U_n^{Rup} e^{-i[\xi_n(x - x_s)]} \tag{10}$$

According to Equation (10), the vibration displacement of mode $n$ at this point in a frequency domain is calculated, and the time domain waveform of the mode is obtained by inverse Fourier transform. To verify the correctness of the method, the following simulations were carried out.

A non-defective three-dimensional rail model with a length of 200 m was established. The vibrations of the railhead along the rail in a longitudinal, transverse, and vertical excitation direction were simulated using ANSYS. To show the results clearly, the excitation signal was selected as a low-frequency signal with a center frequency of 200 Hz and five cycles, which has only four modes. The modal amplitude of each mode at the 200 Hz frequency point after mode analysis is shown in Figure 7.

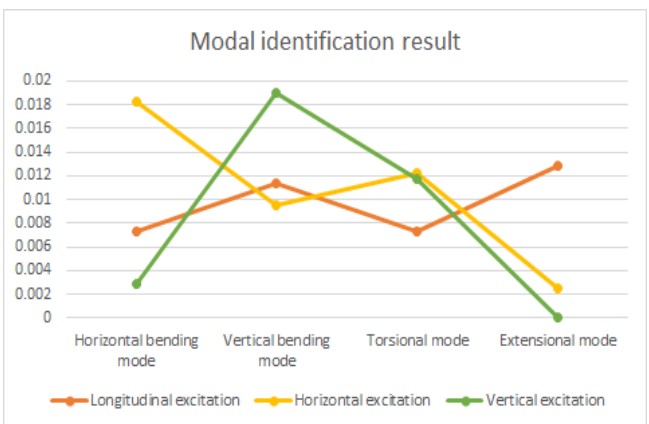

**Figure 7.** Modal identification results under three excitation conditions (200 Hz).

According to Figure 7, when the rail was excited longitudinally, the proportion of the extensional mode was higher. When the rail was excited horizontally, the proportions of the horizontal bending mode and torsion mode were higher. When the rail was excited vertically, the proportions of the vertical bending mode and torsion mode were higher.

Using the SMEA and choosing the frequency range 100–300 Hz, the total waveform at 30 m and four single-mode waveforms were obtained, respectively, as shown in Figure 8a–c.

It can be seen clearly in the figure that only the extensional mode existed when the railhead was excited longitudinally. The horizontal bending and torsion modes existed when the railhead was excited horizontally, and the vertical bending and torsion modes existed when the railhead was excited vertically. The results are consistent with the accurate identification results.

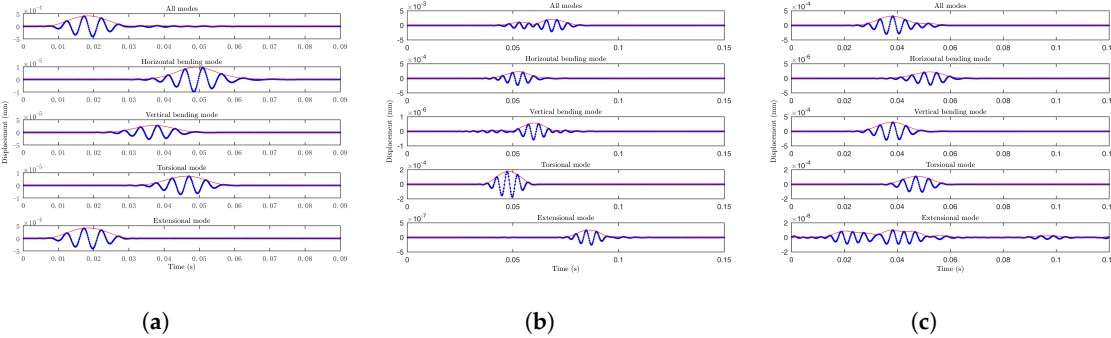

(**a**)                                      (**b**)                                      (**c**)

**Figure 8.** SMEA results: (**a**) longitudinal excitation; (**b**) horizontal excitation; and (**c**) vertical excitation.

## 4. Defect Location

The flow chart of the defect location method is shown in Figure 9.

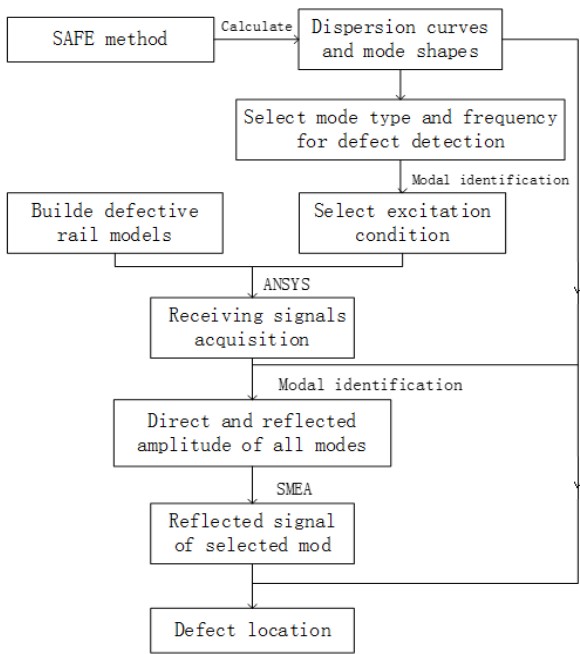

**Figure 9.** Defect location algorithm flow chart.

As can be seen above, the primary task of defect location is to select the mode, frequency, and excitation conditions with which to perform the defect detection. At the same time, a three-dimensional model of the rail with defects must be established. The excitation response of the rail is then simulated by ANSYS, and the vibration displacements of a series of points on the rail are obtained, so as to simulate the signals received by an experiment. According to the results of the modal identification and the SMEA, the reflected signals of the selected modes are plotted. Thereafter, the group velocity and the propagation time of the reflected mode are obtained, so that the defect location can be achieved. The following section takes a transverse crack in a railhead as an example to explain the process of defect location.

### 4.1. Selection of Mode, Frequency, and Excitation Conditions for Defect Detection

The following three principles are used to select the mode, frequency, and excitation conditions for railhead defect detection:

- The mode that only vibrates in the railhead with almost no movement of rail waist and rail bottom and which has a large group velocity is selected.

- The frequency band with better non-dispersive characteristics is selected.
- The mode with the largest amplitude is selected as the excitation condition.

The detection frequency and mode were selected in the frequency range 20–70 kHz. According to Equation (5), the mode shapes of the rail in this frequency range were calculated and drawn. The frequency of 60 kHz was taken as an example, as shown in Figure 10. Among them, the modes in which only the railhead vibrated with almost no movement in the rail waist and rail bottom were modes No. 7 and No. 14.

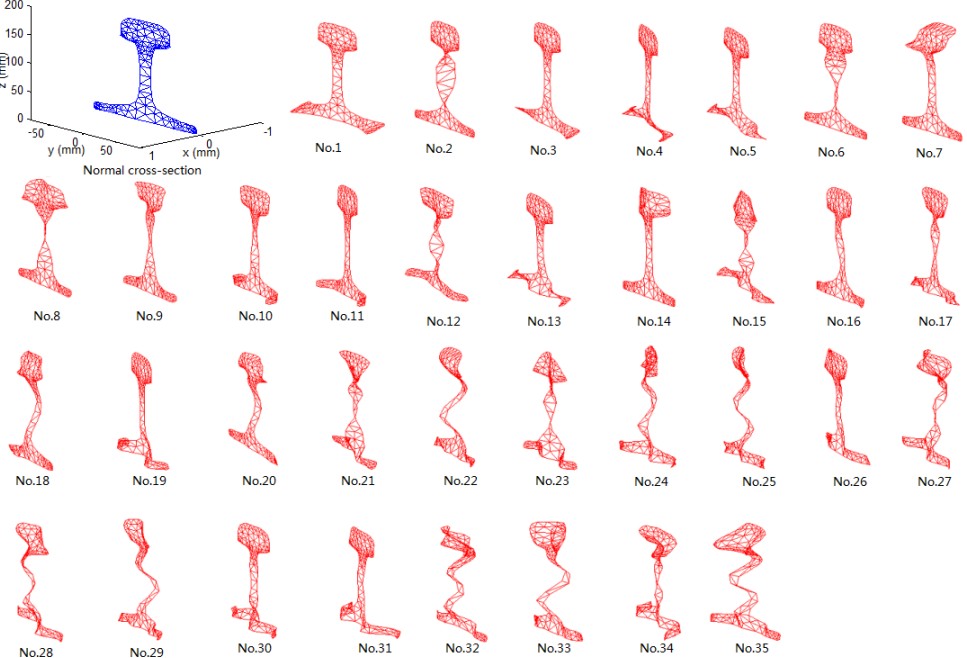

**Figure 10.** Rail mode shapes (60 kHz).

According to Figure 3b, the group velocity dispersion curves of modes No. 7 and No. 14 were extracted, as shown in Figure 11.

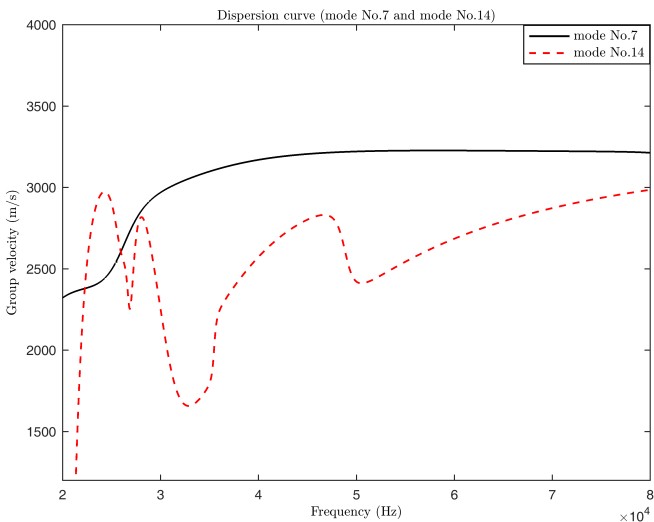

**Figure 11.** Group velocity dispersion curves of modes No. 7 and No. 14.

In Figure 11, we can see that mode No. 7 had almost the same group velocity and excellent non-dispersive characteristics in the proximity of 60 kHz. The group velocity of mode No. 14 was greatly influenced by the frequency and the non-dispersion characteristic was poor. Therefore, mode No. 7 at 60 kHz frequency was selected to detect the defect in the railhead.

A complete three-dimensional rail model with a length of 15 m was established. A simulation whereby the railhead was excited at 7 m along the longitudinal, transverse, and vertical directions by ANSYS was performed. The excitation signal was a sinusoidal wave modulated by Hanning window at the center frequency of 60 kHz with 10 cycles. The co-directional displacement signals between 8 m and 9 m were collected, and the amplitude of each mode under three excitation conditions were calculated using the accurate modal identification method, as shown in Figure 12.

As shown in Figure 12, the highest amplitude for the selected mode is that of vertical excitation. Therefore, vertical excitation was selected.

In summary, mode No. 7 was selected as the detection mode, 60 kHz was selected as the frequency, and vertical excitation was selected as the excitation condition.

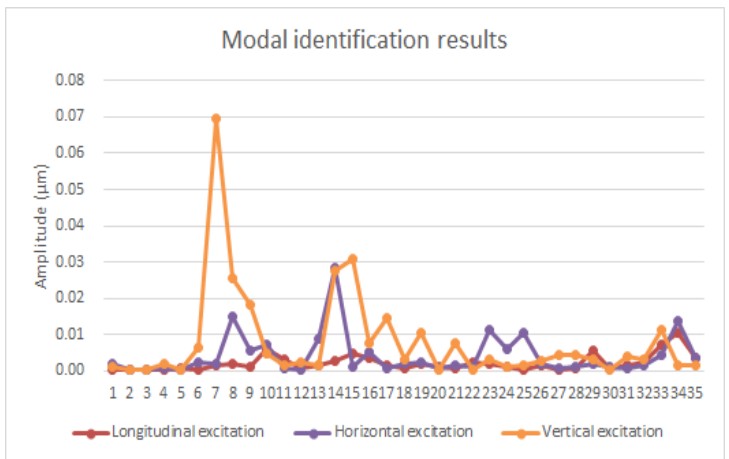

**Figure 12.** Modal identification results under three excitation conditions (60 kHz).

### 4.2. Simulation Analysis of Defect Location

According to the results in Section 4.1, the rail with a defect in the railhead was stimulated. As shown in Figure 13, the rail length was 25 m; $T$ represents the excitation point, which was located at 12 m along the rail. $X$ represents the defect position, and the distance between the excitation point and $X$ was $l = 8$ m. The simulation condition used was the same as the high-frequency excitation presented in Section 2.3.2. Table 2 shows the results of the modal identification.

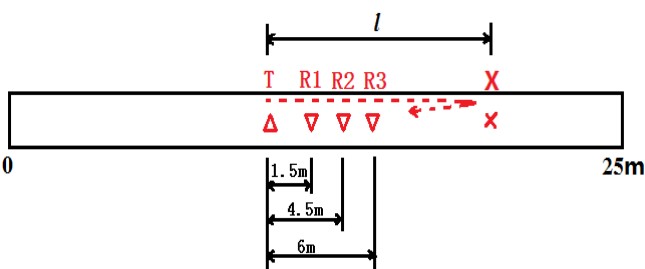

**Figure 13.** Schematic diagram of rail defect location.

The spacings $d$ between the sampling points and the excitation point were 1.5 m, 4.5, m and 6 m, and the collected signal waveforms are shown in Figure 14a–c.

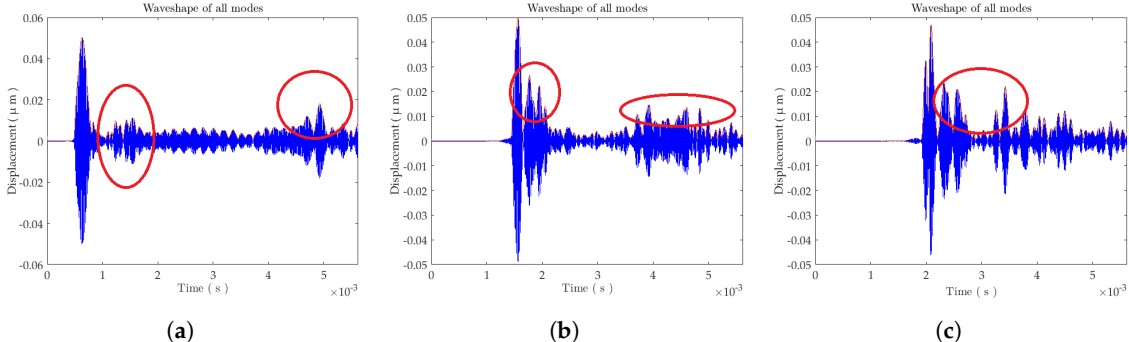

**Figure 14.** Acquisition waveforms for all modes with distances of: (**a**) 1.5 m; (**b**) 4.5 m; and (**c**) 6 m between excitation points and sampling points.

As a result of the complex propagation modes of guided waves in rails, only the first direct wave packet could be distinguished from the waveform. It was impossible to pick out which wave packet in the red circle was the reflected wave packet, and it was also impossible to know for certain which mode or modes each wave packet corresponded to; therefore, it was difficult to give the specific propagation time and the accurate mode group velocity. For this reason, the defect could not be precisely located.

According to the SMEA, the reflected mode waveform of mode No. 7 at 4.5 m from excitation point was calculated in the frequency range 42–78 kHz. As shown in Figure 15b, the peak time $t_1$ of the wave packet was $3.966 \times 10^{-3}$ s. With the peak time $t_0$ of the excitation wave packet ($8.333 \times 10^{-5}$ s) and the group velocity $v_0$ of mode No. 7 (3148.2 m/s), it was possible to calculate the distance between the defect and the excitation point using Equation (11).

$$l = \frac{(t_1 - t_0) * v_0 + d}{2} \tag{11}$$

The actual interval was 8 m and the calculation result was 8.36 m, giving an error of only 0.36 m.

The reflection mode waveforms of mode No. 7 at 1.5 m and 6 m away from the excitation point were extracted, as shown in Figure 15a,c. The distances between the calculated defect and the excitation point were 8.33 m and 8.39 m. Therefore, the errors were 0.33 m and 0.39 m, respectively, both less than 0.5 m, and thus meeting the positioning accuracy requirements.

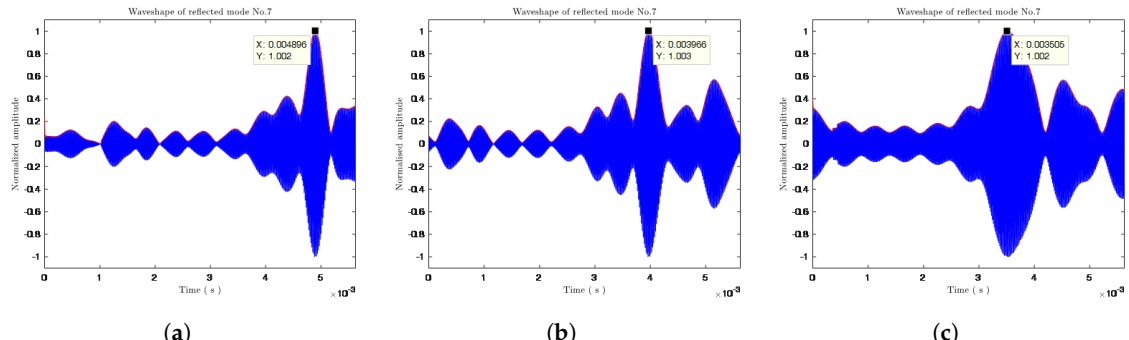

**Figure 15.** The reflection waveforms of mode No. 7 at a distance from the excitation point of: (**a**) 1.5 m; (**b**) 4.5 m; and (**c**) 6 m.

## 5. Conclusions

A rail defect location method based on an SMEA of ultrasonic guided waves was proposed. The dispersion curves of a CHN60 rail were calculated using the semi-analytical finite element method, the mode shapes of the guided waves were calculated, and the actual received signals were predicted using the simulation data provided by ANSYS. Furthermore, accurate modal identification was achieved using the inverse transformation of the excitation response method, and the amplitude of each mode of the guided waves propagating in the rail was obtained. The rail displacement

curves excited by the signals at a low frequency of 200 Hz and high frequency of 60 kHz were correctly predicted.

According to the modal identification results, an SMEA was proposed. The single mode extraction results at 200 Hz were calculated and the results were consistent with those of the modal identification. Thereafter, taking a railhead defect as an example, according to certain selection principles, mode No. 7 with a signal frequency of 60 kHz and a vertical excitation condition were selected. The reflective modes of mode No. 7 at 1.5, 4.5, and 6 m from the excitation point of the railhead defect were extracted using the SMEA, and the distance between the rail defect and excitation point was obtained according to the time difference. The errors in location were 0.33, 0.36, and 0.39 m, respectively, which all fall within the 0.5 m set as the detection requirements. Hence, it can be said that the precise determination of the rail defect was accomplished.

**Author Contributions:** Conceptualization, B.X. and X.X.; methodology, X.X. and L.Z.; software, B.X. and H.S.; validation, Z.Y., L.Z. and H.S.; investigation, B.X.; resources, Z.Y.; writing—original draft preparation, B.X.; and writing—review and editing, B.X., X.X. and L.Z.

**Funding:** This research was funded by the National Key Research and Development Program of China (2016YFB1200401)

**Conflicts of Interest:** The authors declare no conflict of interest.

## Abbreviations

The following abbreviations are used in this manuscript:

SAFE    Semi-analytical finite element
SMEA    Single mode extraction algorithm

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
