# Peer review of "Research on a Rail Defect Location Method Based on a Single Mode Extraction Algorithm"

_applsci, doi:10.3390/app9061107_

Round 1

Reviewer 1 Report

Write in text "Equation (x)" instead of "Equation.(x)" and "Figure (x)" instead of "Figure.(x)"

p 3, line 90: give a reference that explains what CHN60 means

p 3, Fig. 1: xyz is not a right hand system. Please correct.

p 6, Fig. 4: only path 3 is defined in the text. Please define the paths 1 and 2

p 7, line 175: 200 Hz is a very low frequency. It is definitely not ultrasonic (should be above 20 kHz). The reviewer has never seen an experiment with such a low frequency guided wave in a rail. The authors have to provide evidence / references that show that such an experiment is feasible. Otherwise the simulation would be purely academic and not of any real-life relevance.

p 7, line 176: please give more details about the crack: length, depth, width

p 7, Table 1: The values of the extensional mode would give a reflection coefficien of 1.2 and not 0.12 as written! There seems to be a printing error. Furthermore the reviewer is very much astonished about the large reflection coefficients, given the smallness of the crack (3 mm). The wavelength of a 200 Hz wave must be very large, much much larger than 3 mm, and hence the reflection of a 3 mm crack would be expected to be negligible.

p 8, line 196: replace 1. by 2.

p 8, Table 2: Again, the reviewer finds the amplitude reflection coefficients to be very large given the smallness of the crack (3 mm). He really doubts whether for e.g. mode 2 half of the wave is reflected back by this tiny crack.

p 12, line 257: replace vibrats by vibrates

p 13, Fig. 10: clarify what quantity exaclty is displayed. Add units of what is displayed.

p 14, Fig. 12: label y-axis and add units (amplitude relative to excitation)

p 15, Fig. 14: displacements reach values up to 50 Micrometers. This is very very large. The reviewer thinks, that these values are orders of magnitudes beyond experimentally accessible values, hence they are unrealistic.

Author Response

Thank you for your comments. The author's responses are attached.

Reviewer 2 Report

The paper discussed the use of SAFE method and single mode extraction algorithm (SMEA) to detect rail defects by simulation. It seems a pure simulation work and no comparison with experiment. However, the work is good and meaningful to those interested in such field.

The major issue of the paper is the English writing. the paper cannot be accepted for publication in its current form. 

Author Response

Thank you for your approval of my manuscript. Because my mother tongue is not English, even if I try my best, English writing ability still needs to be improved. The manuscript has undergone English language editing by MDPI and resubmitted.The attachment is the English editing certificate. The revision has been highlighted in revised version.

Round 2

Reviewer 1 Report

The current manuscript has been considerably improved in quality with respect to the first version. However, there are still two points which should be addressed: The authors have to discuss and put into relation the current work to their own work: a) already published: Xu, X.N.; Zhuang, L.; Xing, B.; Yu, Z.J.; Zhu, L.Q. An Ultrasonic Guided Wave Mode Excitation Method in Rails[J]. IEEE Access. 2018,6: 60414-60428. b) currently submitted to Applied Sciences: An Ultrasonic Guided Wave Mode Selection and Excitation Method in Rail Defect Detection, Shi Hongmei, Zhuang Lu, Xu Xining, Yu Zujun, Zhu Liqiang. And they have to cite these publications. p 12, Fig. 10: Add dimensional units (x, y, z in mm)

Author Response

The manuscript has been revised and the attachment is the responses.

Reviewer 2 Report

All authors are advised to check their manuscript before submission.   

Author Response

Thanks for your advice, and all authors has checked this manuscript.